# Robust saliency maps with distribution-preserving decoys

## Abstract

Saliency methods help to make deep neural network predictions more interpretable by identifying particular features, such as pixels in an image, that contribute most strongly to the network's prediction. Unfortunately, recent evidence suggests that many saliency methods perform poorly when gradients are saturated or in the presence of strong inter-feature dependence or noise injected by an adversarial attack. In this work, we propose a data-driven technique that uses the distribution-preserving decoys to infer robust saliency scores in conjunction with a pre-trained convolutional neural network classifier and any off-the-shelf saliency method. We formulate the generation of decoys as an optimization problem, potentially applicable to any convolutional network architecture. We also propose a novel decoy-enhanced saliency score, which provably compensates for gradient saturation and considers joint activation patterns of pixels in a single-layer convolutional neural network. Empirical results on the ImageNet data set using three different deep neural network architectures—VGGNet, AlexNet and ResNet—show both qualitatively and quantitatively that decoy-enhanced saliency scores outperform raw scores produced by three existing saliency methods.

## 1 Introduction

Deep neural networks (DNNs) deliver remarkable performance in an increasingly wide range of application domains, but they often do so in an inscrutable fashion, delivering predictions without accompanying explanations. In a practical setting such as automated analysis of pathology images, if a patient sample is classified as malignant, then the physician will want to know which parts of the image contribute to this diagnosis. Thus, in general, a DNN that delivers explanations alongside its predictions will enhance the credibility and utility of its predictions for end users (Lipton, 2016).

Two primary classes of methods have been developed for interpreting CNN models. The first class of methods, often referred to as *saliency maps*, use gradients of classification outcome as a per-pixel importance score to characterize the influence of individual pixels to the prediction (Simonyan et al., 2013; Selvaraju et al., 2016; Binder et al., 2016; Shrikumar et al., 2016; Smilkov et al., 2017; Sundararajan et al., 2017; Levine et al., 2019). Despite providing apparently meaningful interpretations in practice, these methods exhibit fundamental limitations. First, saliency maps evaluate pixel-wise importance in an isolated fashion by design, implicitly assuming that other pixels are fixed (Singla et al., 2019). Worse still, the presence of *gradient saturation* often breaks the assumption that important features in general correspond to large gradients (Sundararajan et al., 2016; Shrikumar et al., 2016; Smilkov et al., 2017). Specifically, an important pixel may contribute a strong joint effect on the prediction when considered in conjunction with neighboring pixels. However, that important feature may have a tiny gradient locally, in the sense that its marginal contribution is diminishing due to the flattened output of the layer. Finally, Ghorbani et al. (2017) and Kindermans et al. (2017) systematically revealed the fragility of widely-used saliency methods by showing that even an imperceivable perturbation or simple shift transformation of the input data can lead to a large change in the resulting saliency scores. In light of this observation, a key challenge for any saliency method is ensuring that the saliency scores are robust to gradient saturation and perturbations.

A second class of interpretability methods, often based on *counterfactual*, perturb (*i.e.,* nullifying, blurring, adding noise, or inpainting ) small regions of the image to modify the classifier prediction (Fong & Vedaldi, 2017; Dabkowski & Gal, 2017; Chang et al., 2017; Fan et al., 2017; Chang et al.,

2019; Yousefzadeh & O'Leary, 2019; Goyal et al., 2019). Despite identifying meaningful regions in practice, these methods exhibit several limitations. First, counterfactual-based methods implicitly assume that regions containing the object are the ones most contributed to the prediction (Fan et al., 2017). However, Moosavi-Dezfooli et al. (2017) revealed that counterfactual-based methods are also vulnerable to the adversarial attacks, which force these explanation methods to output non-related backgrounds rather than the meaningful objects as important subregions. In addition, the counterfactual images may be potentially far away from the distribution from which the training samples were drawn (Burns et al., 2019), causing the classifier behavior ill-defined. Though some efforts have been made to train the saliency extractor and the classifier simultaneously (Fan et al., 2017; Zołna et al., 2019), the robustness of the resulting explanations are not guaranteed in most cases (Hendrycks & Dietterich, 2019). In light of this, for any perturbation-based method, a key challenge is ensuring that the perturbations are effective yet preserving the training distribution.

In this work, we propose a data-driven technique to infer robust saliency maps with distribution-preserving perturbations. Given an image of interest, the core idea of this method is to generate a population of perturbed images, referred to as *decoy images*, that resemble the neural network's intermediate representation of the original image but are conditionally independent of its label. The resulting decoy images capture the variation of the image data originating from either sensor noise or adversarial attacks. The major differences between the proposed and counterfactual-based methods are threefold. First, unlike counterfactual images that seek the minimum set of features to exclude so as to minimize the prediction score or to include so as to maximize the prediction score (Fong & Vedaldi, 2017), saliency map methods like the one we propose in this paper aim to characterize the influence of each feature on the prediction score. Second, unlike counterfactual images that are optimized toward the aforementioned objective with respect to the decision boundary, decoys are designed to be independent of labels, without changing the decision boundary at all. Finally, unlike counterfactual images which could potentially be out-of-distribution by adding noise (Smilkov et al., 2017), rescaling (Sundararajan et al., 2017), blurring (Fong & Vedaldi, 2017), or inpainting the image (Chang et al., 2017), decoys are plausibly constructed in the sense that their intermediate representations are non-discriminative from the original input data by design.

In addition, we propose a novel decoy-enhanced saliency score, which considers an ensemble of saliency maps across multiple decoy images. We also derive a closed-form formula for the decoy-enhanced saliency score generated from a single-layer convolutional neural network. This analysis shows that the proposed saliency score for each pixel can be decomposed into two components: the range of values for a given pixel among the decoys and the number of neurons jointly activated by that pixel and its neighbors. The former compensates for gradient saturation (Sundararajan et al., 2016) in the sense that an important pixel can vary without strongly affecting the prediction, due to the joint effect of nearby pixels. The latter compensates for the way in which saliency maps treat each pixel independently (Singla et al., 2019) in the sense that the importance of a pixel is jointly determined by the importance of pixels in its vicinity. This closed-form result may provide insights to help design better defensive strategies against adversarial attacks.

We apply the proposed method to the ImageNet dataset (Russakovsky et al., 2015) in conjunction with three standard saliency methods. We demonstrate empirically that decoy-enhanced saliency scores perform better than the original saliency scores, both qualitatively and quantitatively, even in the presence of various adversarial perturbations to the image.

## 2 PROBLEM SETTING

Consider a multi-label classification task in which a pre-trained neural network model implements a function $F: \mathbb{R}^d \mapsto \mathbb{R}^C$ that maps from the given input $\mathbf{x} \in \mathbb{R}^d$ to $C$ predicted classes. The score for each class $c \in \{1, \cdots, C\}$ is $F^c(\mathbf{x})$, and the predicted class is the one with maximum score, *i.e.,* $\arg \max_{c \in \{1, \cdots, C\}} F^c(\mathbf{x})$.

A common instance of this setting is image classification, in which case the input $\mathbf{x}$ corresponds to the pixels of an image. A *saliency method* aims to find the subset of features in the input $\mathbf{x}$ that most strongly lead the neural network to make its prediction. Thus, each pixel is assigned a *saliency score*, encoded in an explanation map $E: \mathbb{R}^d \mapsto \mathbb{R}^d$, in which the pixels with higher scores represent higher "importance" to the final prediction.

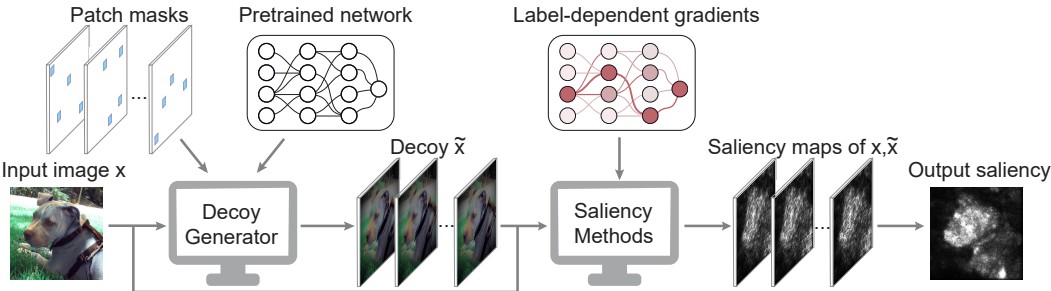

Figure 1: Workflow for creating decoy-enhanced saliency maps.

A variety of saliency methods have been proposed in the literature. Some, such as edge or other generic feature detectors (Adebayo et al., 2018), are independent of the predictive model. In this work, we focus on three methods that do depend on the predictor. The vanilla gradient method (Simonyan et al., 2013) simply calculates the gradient of the class score with respect to the input $\mathbf{x}$, which is defined as $E_{grad}(\mathbf{x}) = \nabla_{\mathbf{x}} F^c(\mathbf{x})$. The SmoothGrad method (Smilkov et al., 2017) seeks to reduce noise in the saliency map by averaging over explanations of noisy copies of an input, defined as $E_{sg}(\mathbf{x}) = \frac{1}{N} \sum_{i=1}^{N} E_{grad}(\mathbf{x}+g_i)$ with noise vectors $g_i \sim N(0, \sigma^2)$. The integrated gradients method (Sundararajan et al., 2017) aims to tackle the problem of gradient saturation. The method starts from a baseline input $\mathbf{x}^0$ and sums over the gradient with respect to scaled versions of the input ranging from the baseline to the observed input, defined as $E_{ig}(\mathbf{x}) = (\mathbf{x} - \mathbf{x}^0) \times \int_0^1 \nabla_{\mathbf{x}} F^c(\mathbf{x}^0 + \alpha(\mathbf{x} - \mathbf{x}^0))d\alpha$.

In this setting, we are given three inputs: a pre-trained neural network model $F$ with $L$ layers, an image of interest $\mathbf{x}$, and a saliency method $E$ such that $E(\mathbf{x}; F)$ is a saliency map of the same dimensions as $\mathbf{x}$. As illustrated in Figure 1, the robust saliency map can be obtained in two steps: 1) constructing decoy images, and 2) computing decoy-enhanced saliency scores.

## 3 DEFINITION OF DECOYS

Say that $F_\ell : \mathbb{R}^d \mapsto \mathbb{R}^{d_\ell}$ is the function instantiated by the given network, which maps from an input image $\mathbf{x} \in \mathbb{R}^d$ to its intermediate representation $F_\ell(\mathbf{x}) \in \mathbb{R}^{d_\ell}$ at layer $\ell \in \{1, 2, \cdots, L\}$. At a specified layer $\ell$, the random vector $\tilde{\mathbf{x}} \in \mathbb{R}^d$ is said to be a *decoy* of $\mathbf{x} \in \mathbb{R}^d$ if the following swappable condition is satisfied:

$$F_\ell(\mathbf{x}) = F_\ell(\mathbf{x}_{\mathrm{swap}(\tilde{\mathbf{x}}, \mathcal{P})}), \text{for image patches } \mathcal{P} \subset \{1, \cdots, d\}, \tag{1}$$

The swap$(\tilde{x}, \mathcal{P})$ operation swaps image patches between $\mathbf{x}$ and $\tilde{\mathbf{x}}$ (Figure 2). Hence, if $\mathcal{P} = \{10\}$, then $x_{\mathrm{swap}(\tilde{\mathbf{x}}, \mathcal{P})}$ is a new image which is almost equivalent to $\mathbf{x}$ except that the 10th patch is exchanged from $\tilde{\mathbf{x}}$. A "valid" patch is a local region receptive to the convolutional filter of a given CNN, for example, the $3 \times 3$ area in VGGNet (Simonyan & Zisserman, 2014). The swappable condition ensures that the original image $\mathbf{x}$ and its decoy $\tilde{\mathbf{x}}$ are indistinguishable in terms of the intermediate representation at layer $\ell$. Note in particular that the construction of decoys relies solely on the first $\ell$ layers of the neural network $F_1, F_2, \cdots, F_\ell$ and is independent of the succeeding layers $F_{\ell+1}, \cdots, F_L$. In other words, $\tilde{\mathbf{x}}$ is conditionally independent of the classification task $F(\mathbf{x})$ given the input $\mathbf{x}$; *i.e.,* $\tilde{\mathbf{x}} \perp\!\!\!\perp F(\mathbf{x})|\mathbf{x}$. See Supplementary Section A6 for the explanation why decoys exist.

Figure 2: An illustrative example of the swap operator swapping image patches between original and decoy images.

### 3.1 GENERATE DECOYS BY OPTIMIZATION

Given an image of interest $\mathbf{x} \in \mathbb{R}^d$ and a patch mask $\mathcal{M} \in \{0, 1\}^d$, we want to construct a decoy image $\tilde{\mathbf{x}} = G_\ell(\mathbf{x}, \mathcal{M}) \in \mathbb{R}^d$, with respect to a specified neural network layer $\ell$, so that $\tilde{\mathbf{x}}$ is as different as possible from $\mathbf{x}$. For this purpose, we optimize the following objective function:

$$\text{maximize}_{\tilde{\mathbf{x}}} \quad \left\| ((\tilde{\mathbf{x}} - \mathbf{x}) \cdot s)^+ \right\|_1,$$
$$\text{s.t. } \|F_\ell(\tilde{\mathbf{x}}) - F_\ell(\mathbf{x})\|_\infty \le \epsilon, \ (\tilde{\mathbf{x}} - \mathbf{x}) \circ (1 - \mathcal{M}) = 0, \text{ and } \tilde{\mathbf{x}} \in [0, 1]^d. \tag{2}$$

where $(\cdot)^+ = \max(\cdot, 0)$, and $s = 1$ or $s = -1$ depends on whether we intend to investigate the upward or downward limit of $\tilde{\mathbf{x}}$ deviating from $\mathbf{x}$, respectively. The operators $\|\cdot\|_1$ and $\|\cdot\|_\infty$ correspond to the $L_1$ and $L_\infty$ norms, respectively. The constraint $(\tilde{\mathbf{x}} - \mathbf{x}) \circ (1 - \mathcal{M}) = 0$ ensures that $\tilde{\mathbf{x}}$ and $\mathbf{x}$ differ only in the area marked by the patch mask $\mathcal{M}$, where $\circ$ denotes entry-wise multiplication. The final constraint ensures that the values in $\tilde{\mathbf{x}}$ fall into an appropriate range for image pixels, $[0, 1]$ in the normalized image case. In principle, the formulation in Equation 2 is applicable to any convolutional neural network architecture ranging from simple convolutional networks to more sophisticated ones (*e.g.*, , AlexNet (Krizhevsky et al., 2012), Inception network (Szegedy et al., 2016) and ResNet (He et al., 2016), etc.).

As illustrated in Figure 1, we can construct a population of $n$ independently-sampled patch masks $\{\mathcal{M}^1, \mathcal{M}^2, \cdots, \mathcal{M}^n\}$, so that each possible image patch is covered by at least one patch mask. Thus we can construct a population of $n$ decoy images $\{\tilde{\mathbf{x}}^1, \tilde{\mathbf{x}}^2, \cdots, \tilde{\mathbf{x}}^n\}$ with respect to different patch masks. Note that the input image $\mathbf{x}$ is also a trivially valid decoy, by definition; *i.e.*, it satisfies Equation 1 and the conditional independence requirement (*i.e.*, $\tilde{\mathbf{x}} \perp\!\!\!\perp F(\mathbf{x})|\mathbf{x}$).

## 3.2 Decoy-enhanced saliency scores

Given the generated population of decoy images $\{\tilde{\mathbf{x}}^1, \tilde{\mathbf{x}}^2, \cdots, \tilde{\mathbf{x}}^n\}$, the given saliency method $E$ can be applied to each decoy image, following the fundamental rule that the saliency method should be agnostic to information revealing which image is a decoy and which is not. In this way, we obtain a corresponding population of decoy saliency maps $\{E(\tilde{\mathbf{x}}^1; F), E(\tilde{\mathbf{x}}^2; F), \cdots, E(\tilde{\mathbf{x}}^n; F)\}$.

Now the pixel-wise variation for each feature $\mathbf{x}_j$ can be characterized by a population of feature values $\tilde{X}_j = \{\tilde{\mathbf{x}}_j^1, \tilde{\mathbf{x}}_j^2, \cdots, \tilde{\mathbf{x}}_j^n\}$. Accordingly, instead of merely quantifying the saliency of each feature $\mathbf{x}_j$ via a single value $E(\mathbf{x}; F)_j$, we can now characterize the uncertainty of the feature's saliency via the corresponding population of saliency scores $\tilde{E}_j = \{E(\tilde{\mathbf{x}}^1; F)_j, E(\tilde{\mathbf{x}}^2; F)_j, \cdots, E(\tilde{\mathbf{x}}^n; F)_j)\}$.

We define the decoy-enhanced saliency score $Z_j$ for each feature $\mathbf{x}_j$ as

$$Z_j = \max(\tilde{E}_j) - \min(\tilde{E}_j), \tag{3}$$

That is, $Z_j$ is determined by the empirical variation of the decoy's saliency scores.

## 3.3 Example: single-layer CNN

Consider a single-layer convolutional neural network with decoy swappable patch size $1 \times 1$ and convolutional size $3 \times 3$. The input of this CNN is $\mathbf{x} \in \mathbb{R}^d$, unrolled from a $\sqrt{d} \times \sqrt{d}$ grayscale image matrix. Similarly, we have a $3 \times 3$ convolutional filter unrolled into $\mathbf{g} \in \mathbb{R}^9$, where $\mathbf{g}$ is indexed as $\mathbf{g}_k$ for $k \in \mathcal{K}$, here $\mathcal{K} = \{-\sqrt{d} - 1, -\sqrt{d}, -\sqrt{d} + 1, -1, 0, 1, \sqrt{d} - 1, \sqrt{d}, \sqrt{d} + 1\}$, corresponding to the index shift in matrix form from the top-left to bottom-right pixel, respectively. We denote $(\mathbf{g}*\mathbf{x}) \in \mathbb{R}^d$ as the output of the convolution operation on the input $\mathbf{x}$, where $(\mathbf{g}*\mathbf{x})_i = \sum_{k \in \mathcal{K}} \mathbf{g}_k \mathbf{x}_{i+k}$. For simplicity, we further assume that there are no pathological cases such as $(\mathbf{g} * \mathbf{x})_i = 0$. Such a neural network can be represented as

$$\mathbf{m} = \text{relu}(\mathbf{g} * \mathbf{x}), \quad \mathbf{o} = \mathbf{W}^T\mathbf{m} + \mathbf{b}, \quad \mathbf{p} = \text{softmax}(\mathbf{o}), \tag{4}$$

where $\text{relu}(\cdot)$ is the entry-wise ReLU operator (Glorot et al., 2011), $\mathbf{W} \in \mathbb{R}^{d \times C}$ represents the combined weights of the neural network, and $\mathbf{b} \in \mathbb{R}^C$ represents the biases. The terms $\mathbf{o} \in \mathbb{R}^C$ and $\mathbf{p} \in \mathbb{R}^C$ are the logits and the predicted class probabilities, respectively. The entry-wise softmax operator for target class $c$ is defined as $\mathbf{p}_c = \frac{e^{\mathbf{o}_c}}{\sum_{c'=1}^C e^{\mathbf{o}_{c'}}}$, for $c \in \{1, 2, \cdots, C\}$.

In this case, we have the following result:

**Proposition 1.** $Z_j$ *is bounded by:*

$$Z_j \leq C_2 \left|(\tilde{\mathbf{x}} - \mathbf{x})_j\right| \sum_{k \in \mathcal{K}} \tilde{a}_{j+k} + C_1, \tag{5}$$

where $\tilde{\mathbf{x}}$ is the decoy which maximizes $E(\tilde{\mathbf{x}}; F)_j - E(\mathbf{x}; F)_j$, $\tilde{a}_j = \mathbf{1}\{(\mathbf{g} * \tilde{\mathbf{x}})_j \geq 0\}$, and $C_1 > 0$ and $C_2 > 0$ are bounded constants. See Supplementary Section A7 for the full proof.

Proposition 1 indicates that the saliency statistic is determined by two factors: the range of values associated with pixel $j$ among the decoys ($|(\tilde{\mathbf{x}} - \mathbf{x})_j|$) and the number of neurons jointly activated by pixel $j$ and its neighbors ($\sum_{k \in \mathcal{K}} \tilde{a}_{j+k}$). The former explains the gradient saturation problem (Sundararajan et al., 2016) in the sense that important features may have more room to fluctuate without influencing the joint effect on the prediction. The latter compensates for way in which saliency maps treat each pixel independently (Singla et al., 2019) in the sense that the importance of a pixel $\mathbf{x}_j$ is jointly determined by the surrounding pixels (*i.e.*, the $3 \times 3$ localized region in our setting), potentially capturing meaningful patterns such as edges, texture, etc. In this way, the proposed saliency statistic not only compensates for gradient saturation but takes joint activation patterns into consideration as well. Indeed, the bound provided by Proposition 1 can be considered a motivation for defining the decoy-enhanced saliency score as we have in Equation 3.

### 3.4 IMPLEMENTATION DETAILS

We solve Equation 2 by augmenting a constraint as a penalty in the objective function:

$$\text{minimize}_{\tilde{\mathbf{x}}} \quad -\left\|((\tilde{\mathbf{x}} - \mathbf{x}) \cdot s)^+\right\|_1 + c \cdot \left\|F_\ell(\tilde{\mathbf{x}}) - F_\ell(\mathbf{x})\right\|_\infty,$$
$$\text{s.t.} \quad (\tilde{\mathbf{x}} - \mathbf{x}) \circ (1 - \mathcal{M}) = 0, \text{ and } \tilde{\mathbf{x}} \in [0, 1]^d. \tag{6}$$

When the constant $c > 0$ is properly chosen, Equations 2 and 6 are equivalent in the sense that the optimal solution to Equation 2 equals the optimal solution to Equation 6. Our strategy is to set $c$ to a small value initially and run the optimization. If it fails, then we double $c$ and repeat until success. The sensitivity of choosing $c$ is reported in Supplementary Section A9.

To eliminate the constraint on pixel values ($\tilde{\mathbf{x}} \in [0, 1]^d$), we use the change-of-variable trick (Carlini & Wagner, 2017). Note that other transformations are possible yet unexplored in this paper. Specifically, instead of optimizing over $\tilde{\mathbf{x}}$, we introduce $\hat{\mathbf{x}}$ satisfying $\tilde{\mathbf{x}}_i = \frac{1}{2}(\tanh(\hat{\mathbf{x}}_i) + 1)$, for all $i \in \{1, 2, \cdots, d\}$. Because $\tanh(\hat{\mathbf{x}}_i) \in [-1, 1]$ implies $\tilde{\mathbf{x}}_i \in [0, 1]$, any solution to $\hat{\mathbf{x}}$ is naturally valid.

To eliminate of the patch mask constraint ($(\tilde{\mathbf{x}} - \mathbf{x}) \circ (1 - \mathcal{M}) = 0$), we use projected gradient descent during the optimization stage. Specifically, after performing each step of standard gradient descent, we enforce $\tilde{\mathbf{x}} = \tilde{\mathbf{x}} \circ \mathcal{M} + \mathbf{x} \circ (1 - \mathcal{M})$.

Putting these ideas together, we minimize the following objective function:

$$\text{minimize}_{\hat{\mathbf{x}}} \quad -\left\|((\text{arctanh}(2\hat{\mathbf{x}} - 1) - \mathbf{x}) \cdot s)^+\right\|_1 + c \cdot \left\|F_\ell(\text{arctanh}(2\hat{\mathbf{x}} - 1)) - F_\ell(\mathbf{x})\right\|_\infty, \tag{7}$$

Because the $L_\infty$ norm is not fully differentiable, we adopt the trick introduced by Carlini & Wagner (2017). Specifically, in each iteration we solve:

$$\text{minimize}_{\hat{\mathbf{x}}} -\left\|\max((\text{arctanh}(2\hat{\mathbf{x}} - 1) - \mathbf{x}) \cdot s, 0)\right\|_1 + c \cdot \left\|(|F_\ell(\text{arctanh}(2\hat{\mathbf{x}} - 1)) - F_\ell(\mathbf{x})| - \tau)^+\right\|_2^2, \tag{8}$$

where $\tau > 0$ is initialized with a large value (1 in our experiment). After each iteration, if the second term in Equation 8 is zero, indicating that $\tau$ is too large, then we reduce $\tau$ by a factor of $0.95$ and repeat; otherwise, we terminate the optimization. The run time of optimization is reported in Supplementary Section A10.

## 4 EXPERIMENTS

To evaluate the effectiveness of our proposed method, we perform extensive experiments on deep learning models that target image classification tasks. The performance of our method is assessed both qualitatively and quantitatively. The results show that our proposed method identifies intuitively more coherent saliency maps than existing state-of-the-art saliency methods alone. The method also achieves quantitatively better alignment to truly important features and demonstrates stronger robustness to adversarial manipulation.

Our experiments primarily use the VGG16 model (Simonyan & Zisserman, 2014) pretrained on the ImageNet dataset (Russakovsky et al., 2015). We also demonstrate the applicability of our method to other well-studied CNN architectures such as AlexNet (Krizhevsky et al., 2012) and ResNet (He et al., 2016). The settings for all experiments are reported in Supplementary Section A13.

SF: 5.74  SF: -0.26  SF: -0.26  SF: -0.11  SF: 3.21  SF: -0.36  SF: -0.26  SF: -0.09  SF: -0.07  SF: -0.48  SF: -0.43  SF: -0.11

Figure 3: Visualization of saliency maps comparing three state-of-the-art saliency methods with and without decoy images generated from three techniques (*i.e.,* our decoy generation method, blurring, and inpainting). More examples can be found in Supplementary Section A14.

### 4.1 BENCHMARKS AND VISUALIZATION METHODOLOGIES

To benchmark the performance of our proposed method, we considered the following baseline saliency methods: the vanilla gradient method (Simonyan et al., 2013), SmoothGrad (Smilkov et al., 2017) and integrated gradients (Sundararajan et al., 2017).

To evaluate our decoy generation method, we used the following two methods to replace the proposed decoy generation method: blurring and inpainting. In particular, given an image with a few masked patches (*i.e.,* $\mathbf{x} \circ (1 - \mathcal{M})$), instead of generating a decoy image (*i.e.,* $\tilde{\mathbf{x}}$) by solving the optimization function in Equation 2, we fill the missing parts by replacing the original missing patches with blurring or using the state-of-the-art inpainting method (Yu et al., 2018).[1] Using the decoy images generated by these two substitute methods, we then computed the decoy-enhanced saliency scores for the three baseline saliency methods. We compared the results of blurring and inpainting with our method, both qualitatively and quantitatively.

A heatmap is a typical way of displaying a saliency map. Our strategy for projecting a saliency map onto a heatmap is as follows. Saliency methods produce signed values for input features (pixels in our experiment), where the sign of the value indicates the direction of influence for the corresponding feature to the predicted class. We first took the absolute value of the saliency maps, as suggested by Smilkov et al. (2017). Then we used the maximum value across all color channels, as suggested by Simonyan et al. (2013), to arrive at a single saliency value for each pixel. To avoid outlier pixels with extremely high saliency values leading to an almost entirely black heatmap, we winsorized those outlier saliency values to a relatively high value (*e.g.,* $95^{th}$ percentile), as suggested by Smilkov et al. (2017), to achieve better visibility. Finally, the values are linearly rescaled to the range $[0, 1]$.

### 4.2 COMPARISON CRITERIA

To comprehensively evaluate our proposed method against the aforementioned baseline saliency methods, we focus on the following three goals.

First, we aim to achieve visual coherence of the identified saliency map. Intuitively, we would prefer a saliency method that produces a saliency map that aligns cleanly with the object of interest.

Second, we use a saliency fidelity metric (Dabkowski & Gal, 2017) that quantifies the influence of important features identified by a saliency method, defined as $S_F(E(\mathbf{x}; F), \mathbf{x}, c) = -(\log(F^c(E(\mathbf{x}; F) \circ \mathbf{x})) - \log(F^c(E(\mathbf{x}; F)_{mean} \circ \mathbf{x})))$, where $c$ indicates the predicted class of input $\mathbf{x}$ and the saliency map $E(\mathbf{x}; F)$ is normalized into a $[0, 1]$ range, as described in Section 4.1. The function $F^c(E(\mathbf{x}; F) \circ \mathbf{x})$ performs entry-wise multiplication between $E(\mathbf{x}; F)$ and $\mathbf{x}$, encoding the overlaps between the object of interest and the concentration of information by the saliency map. The rationale here is that, by viewing the saliency score of the feature as its contribution to the predicted class, a good saliency method will be able to weight important features more highly than less important ones, giving rise to better predicted class scores. Accordingly, a lower value for the saliency fidelity metric implies a faithful saliency method. Note that we subtract the influence of mean saliency $E(\mathbf{x}; F)_{mean}$ (*i.e.,* . the mean of $E(\mathbf{x}; F)$ on each channel) to eliminate the influence of bias in $E(\mathbf{x}; F)$. As a result, we can exclude some trivial cases (*e.g.,* $E(\mathbf{x}; F) = \mathbf{1}$).

Third, we aim for robustness to adversarial manipulations (Ghorbani et al., 2017). Adversarial attacks aim to apply imperceptible perturbations to an input image that do not affect the predicted class but produce a very different saliency map relative to the map from the unperturbed image. Ghorbani et al.

---

[1]We directly used the well-trained inpainting model provide by Yu et al. (2018).

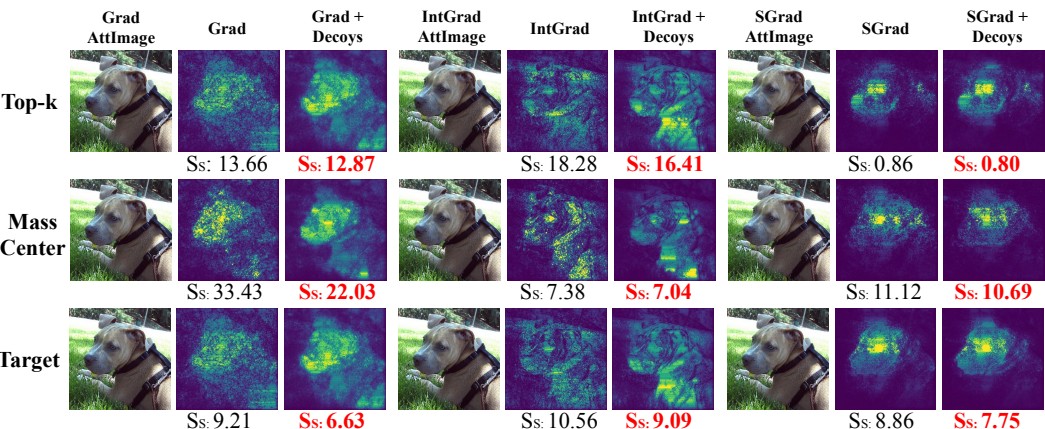

Figure 4: Visualization of saliency maps under adversarial attacks. More examples can be found in Supplementary Section A14.

(2017) propose three different attacks: (1) the top-k attack seeks to decrease the important scores of the top-k important features; (2) the target attack aims to increase the feature importance of a pre-specified region in the input image; (3) the mass-center attack aims to spatially change the center of mass of the original saliency map. In this paper, we specify the bottom-right $4 \times 4$ region of the original image for the target attack and select $k = 5000$ in the top-k attack.

To quantify the robustness of a saliency method $E$ to adversarial attack, we use the sensitivity metric (Alvarez-Melis & Jaakkola, 2018), defined as $S_S(E(\cdot, F), \mathbf{x}, \hat{\mathbf{x}}) = \frac{\|(E(\mathbf{x}, F) - E(\hat{\mathbf{x}}, F))\|_2}{\|\mathbf{x} - \hat{\mathbf{x}}\|_2}$, where $\hat{\mathbf{x}}$ is the perturbed image of $\mathbf{x}$. A small value means that similar inputs do not lead to substantially different saliency maps.

## 4.3 RESULTS

We applied all three state-of-the-art saliency methods and their decoys to a dozen randomly sampled images from ImageNet dataset. A side-by-side comparison of the resulting saliency maps (Figure 3) suggests that decoys consistently help to reduce noise and produce more visually coherent saliency maps. For example, in Figure 3, the gradient method without decoys highlights the region corresponding to the dog's head, but in a scattered format. In contrast, the decoy-enhanced gradient method not only highlights the missing body but identifies the dog head with more details such as ears, cheek, and nose. The visual coherence is also quantitatively supported by the saliency fidelity metric $S_F$, which is consistently higher with decoys than without decoys. The sanity check (Adebayo et al., 2018) of decoy-enhanced saliency method is reported in Supplementary Section A8.

Figure 3 also shows the comparison of our decoy generation method with blurring and inpainting. Our decoy generation method consistently outperforms both competing methods, achieving higher fidelity scores on all of the three baseline saliency methods. The reason is as follows. As is discussed in Proposition 1, to obtain a higher decoy score, a set of decoy images should satisfy the following two requirements: first, they produce similar intermediate representation when provided as input to the target neural network; second, they exhibit large decoy variation. In this paper, we design our decoy generation objective function based on these two requirements. In contrast, neither blurring nor inpainting is designed to meet these requirements. Specifically, blurring may violate the intermediate representation constraint and inpainting may not be able to produce enough decoy variation.

It is worth mentioning that combining image blurring or inpainting with our decoy-enhanced saliency scores can still improve upon the baseline saliency maps. The empirical validity of using image blurring is explained in Supplementary Section A11. From the practitioners' perspective, blurry images can be used as more efficient alternatives to combine with the decoy-enhanced saliency score to achieve comparable performance. However, it is worth mentioning that decoy images are still necessary to justify the theoretical soundness of the decoy-enhanced saliency score.

Next, to test the robustness of the methods we subjected each to the three aforementioned adversarial attacks (Figure 4). Though not fully resistant to adversarial attacks, we observe that enhancing

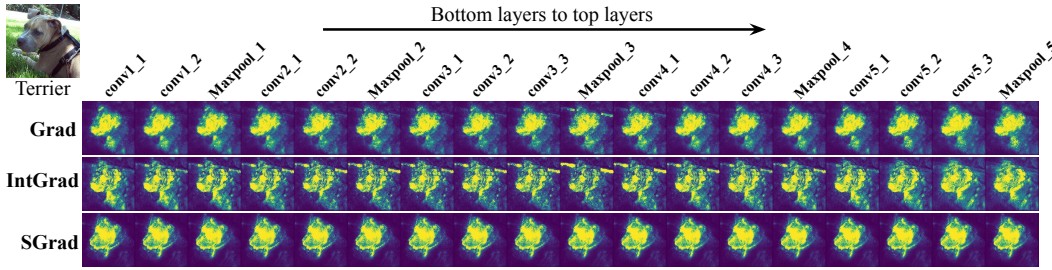

Figure 5: Generating decoys with respect to different DNN layers.

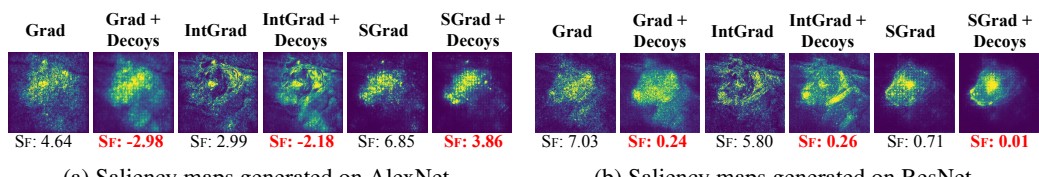

(a) Saliency maps generated on AlexNet.  (b) Saliency maps generated on ResNet.

Figure 6: Visualization of saliency maps under different CNN architectures.

existing saliency methods with decoys consistently mitigates the impact of adversarial manipulations, leading to visually more coherent saliency maps as well as lower sensitivity scores. We believe the extra benefits in the robustness is brought by our decoy-enhanced saliency score. In a normal situation (when the image doesn't suffer from an adversarial attack), an important pixel is not important in an isolated fashion. Instead, the important pixel tends to contribute a strong joint effect in conjunction with neighboring important pixels, to potentially capture meaningful patterns such as edges, texture, etc. In light of this observation, this particular important pixel will have more room to fluctuate without influencing the joint effect on the prediction. In such case, some elements of $\tilde{E}_j$ will be high and others will be low, contributing a large $Z_j$. In the unusual situation when an isolated important pixels is indeed observed (*i.e.,* a pixel is very important for all decoy images and has no room for fluctuation.), we tend to believe that the pixel has been adversarial attacked. In this case, all elements of $\tilde{E}_j$ will be high and the proposed decoy-enhanced saliency score $Z_j$ will be low, which is what we want.

### 4.4 SENSITIVITY OF DECOY SETTINGS

We also conduct experiments to understand the impact of decoy settings on the performance of different saliency methods. Specifically, we focus on the choice of network layer $\ell$, because the intermediate representations can vary significantly at different layers. Accordingly, we vary the value of $\ell$ for VGG16 and compare the differences of decoy-enhanced saliency scores from the three saliency methods, ranging from the first convolutional layer to the last pooling layer. As shown in Figure 5 and Supplementary Section A14, the decoy-enhanced saliency scores generated from different layers for the same image are of similar qualities. We also compute the $S_F$ score for each saliency map. The mean and standard deviation, respectively, for the scores obtained by the gradient, integrated gradient and SmoothGrad methods are as follows: (-0.11, 0.02), (-0.18, 0.02), (-0.21, 0.01). These quantitative results also support the conclusion that our method is not sensitive to layer. This is likely because, as previous research has shown (Chan et al., 2015; Saxe et al., 2011), the final classification results of a DNN are not highly related to the intermediate representations. As a result, decoys for the same sample with the same label from different layers should yield similar results.

### 4.5 APPLICABILITY TO OTHER NEURAL ARCHITECTURES

In addition to the VGG16 model (Simonyan & Zisserman, 2014), we generate saliency maps for other neural architectures: AlexNet (Krizhevsky et al., 2012) and ResNet (He et al., 2016). We visualize their saliency maps in Figure 6 and Supplementary Section A14. From Figure 6, we observe that all the saliency maps accurately highlight the contour of the dog and achieve higher fidelity score

than the baseline saliency methods. This indicates that we can apply our method to various neural architectures and expect consistent performance. It should be noted that, in comparison with saliency maps derived from other neural architectures, the maps tied to AlexNet are relatively vague. We hypothesize that this is due to the relative simplicity of the AlexNet architecture.

## 5 DISCUSSION AND CONCLUSION

In this work, we have used distribution-preserving decoys as a way to produce more robust saliency scores. First, we formulate decoy generation as an optimization problem, in principle applicable to any network architecture. We demonstrate the superior performance of our method relative to three standard saliency methods, both qualitatively and quantitatively, even in the presence of various adversarial perturbations to the image. Second, by deriving a closed-form formula for the decoy-enhanced saliency score, we show that our saliency scores compensate for the frequently violated assumption in saliency methods that important features in general correspond to large gradients.

Although we have shown that decoys can be used to produce more robust saliency scores, there remain some interesting directions for future work. For example, one could potentially reframe interpretability as hypothesis testing and use decoys to deliver a set of salient pixels, subject to false discovery rate control at some pre-specified level (Burns et al., 2019; Lu et al., 2018).

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

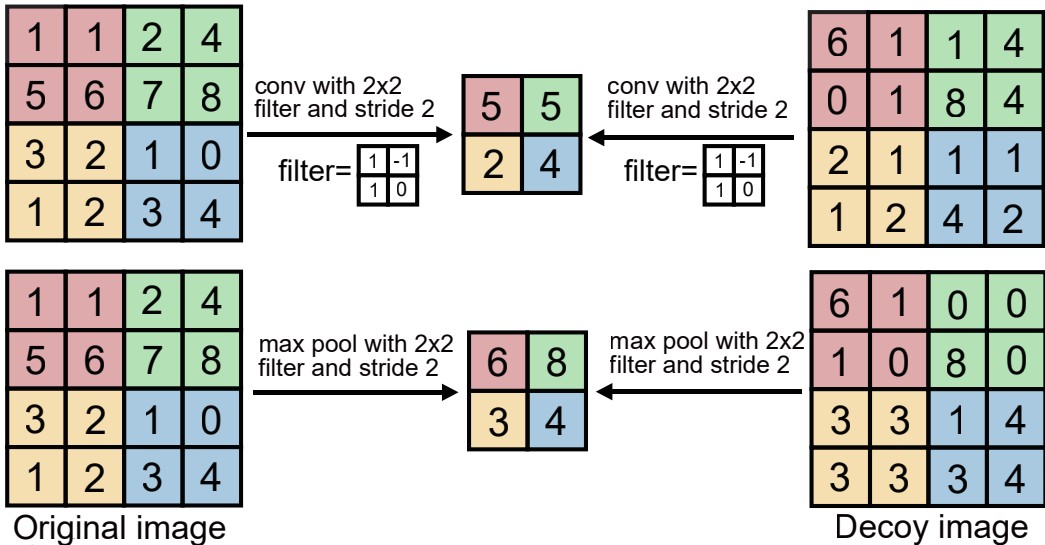

Figure A7: A toy example to illustrate why decoys exist.

Mukund Sundararajan, Ankur Taly, and Qiqi Yan. Gradients of counterfactuals. *arXiv:1611.02639*, 2016.

Mukund Sundararajan, Ankur Taly, and Qiqi Yan. Axiomatic attribution for deep networks. *arXiv:1703.01365*, 2017.

Christian Szegedy, Wojciech Zaremba, Ilya Sutskever, Joan Bruna, Dumitru Erhan, Ian Goodfellow, and Rob Fergus. Intriguing properties of neural networks. *arXiv:1312.6199*, 2013.

Christian Szegedy, Vincent Vanhoucke, Sergey Ioffe, Jon Shlens, and Zbigniew Wojna. Rethinking the inception architecture for computer vision. In *Proceedings of the IEEE conference on Computer Vision and Pattern Recognition*, pp. 2818–2826, 2016.

Roozbeh Yousefzadeh and Dianne P O'Leary. Interpreting neural networks using flip points. *arXiv preprint arXiv:1903.08789*, 2019.

Jiahui Yu, Zhe Lin, Jimei Yang, Xiaohui Shen, Xin Lu, and Thomas S Huang. Generative image inpainting with contextual attention. In *Proceedings of the IEEE Conference on Computer Vision and Pattern Recognition*, pp. 5505–5514, 2018.

Konrad Zołna, Krzysztof J Geras, and Kyunghyun Cho. Classifier-agnostic saliency map extraction. In *Proceedings of the AAAI Conference on Artificial Intelligence*, volume 33, pp. 10087–10088, 2019.

## A6  EXISTENCE OF DECOYS

As illustrated in Figure A7, the original image and the decoy image shares exactly the same intermediate representation with respect to convolutional and max pooling.

## A7  PROOF OF PROPOSITION 1

Consider a single-layer convolutional neural network with decoy swappable patch size $1 \times 1$ and convolutional size $3 \times 3$. The input of this CNN is $\mathbf{x} \in \mathbb{R}^d$, unrolled from a $\sqrt{d} \times \sqrt{d}$ grayscale image matrix. Similarly, we have a $3 \times 3$ convolutional filter unrolled into $\mathbf{g} \in \mathbb{R}^9$, where $\mathbf{g}$ is indexed as $\mathbf{g}_k$ for $k \in \mathcal{K}$, here $\mathcal{K} = \left\{ -\sqrt{d} - 1, -\sqrt{d}, -\sqrt{d} + 1, -1, 0, 1, \sqrt{d} - 1, \sqrt{d}, \sqrt{d} + 1 \right\}$, corresponding

to the index shift in matrix form from the top-left to bottom-right pixel, respectively. We denote $(\mathbf{g}*\mathbf{x}) \in \mathbb{R}^d$ as the output of the convolution operation on the input $\mathbf{x}$, where $(\mathbf{g}*\mathbf{x})_i = \sum_{k \in \mathcal{K}} \mathbf{g}_k \mathbf{x}_{i+k}$. For simplicity, we further assume that there are no pathologic cases such as $(\mathbf{g} * \mathbf{x})_i = 0$. Such a neural network can be represented as

$$\begin{aligned}
\mathbf{m} &= \text{relu}(\mathbf{g} * \mathbf{x})\,, \\
\mathbf{o} &= \mathbf{W}^T\mathbf{m} + \mathbf{b}\,, \\
\mathbf{p} &= \text{softmax}(\mathbf{o})\,,
\end{aligned} \tag{9}$$

where $\text{relu}(\cdot)$ is the entry-wise ReLU operator, $\mathbf{W} \in \mathbb{R}^{d \times C}$ represents the combined weights of the neural network, and $\mathbf{b} \in \mathbb{R}^C$ represents the biases. The terms $\mathbf{o} \in \mathbb{R}^C$ and $\mathbf{p} \in \mathbb{R}^C$ are the logits and the predicted class probabilities, respectively. The entry-wise softmax operator for target class $c$ is defined as $\mathbf{p}_c = \frac{e^{\mathbf{o}_c}}{\sum_{c'=1}^{C} e^{\mathbf{o}_{c'}}}$, for $c \in \{1, 2, \cdots, C\}$.

The gradient of $\mathbf{p}_c$ with respct to $\mathbf{x}$ can be written as follows, using the denominator layout notation of the derivative of a vector:

$$\nabla_{\mathbf{x}}\mathbf{p}_c = \frac{\partial \mathbf{m}}{\partial \mathbf{x}} \frac{\partial \mathbf{o}}{\partial \mathbf{m}} \frac{\partial \mathbf{p}_c}{\partial \mathbf{o}}\,, \tag{10}$$

where

$$\frac{\partial \mathbf{o}}{\partial \mathbf{m}} = \mathbf{W}\,, \tag{11}$$

and

$$\begin{cases}
\frac{\partial \mathbf{p}_c}{\partial \mathbf{o}_{c'}} = (\mathbf{p}_c - \mathbf{p}_c^2) & \text{if } c' = c\,, \\
\frac{\partial \mathbf{p}_c}{\partial \mathbf{o}_{c'}} = -\mathbf{p}_c\mathbf{p}_{c'} & \text{otherwise}\,.
\end{cases} \tag{12}$$

Then, we can write $\frac{\partial \mathbf{p}_c}{\partial \mathbf{o}_{c'}}$ as follows:

$$\frac{\partial \mathbf{p}_c}{\partial \mathbf{o}_{c'}} = \hat{\mathbf{P}}_{\cdot c}\,, \tag{13}$$

where $\hat{\mathbf{P}}_{\cdot c}$ corresponds to the $c$-th column of $\hat{\mathbf{P}}$ and $\hat{P} = \text{diag}(\mathbf{p}) - \mathbf{p}\mathbf{p}^T$.

We then define $\frac{\partial \mathbf{m}}{\partial \mathbf{x}}$ as $\mathbf{B} \in \mathbb{R}^{d \times d}$, in which

$$\mathbf{B}_{ij} = \frac{\partial \mathbf{m}_j}{\partial \mathbf{x}_i} = \frac{\partial (\mathbf{g} * \mathbf{x})_j}{\partial \mathbf{x}_i} \frac{\partial \mathbf{m}_j}{\partial (\mathbf{g} * \mathbf{x})_j} = a_j \frac{\partial (\mathbf{g} * \mathbf{x})_j}{\partial \mathbf{x}_i}\,, \tag{14}$$

where $a_j$ is denoted as $a_j = \mathbf{1}\{(\mathbf{g} * \mathbf{x})_j \geq 0\}$. Then, we have

$$\begin{cases}
\mathbf{B}_{ij} = a_j\mathbf{g}_{i-j} & \text{if } i - j \in \mathcal{K}\,, \\
\mathbf{B}_{ij} = 0 & \text{otherwise}\,.
\end{cases} \tag{15}$$

Given Equations 11, 13 and 15, Equation 10 can be rewritten as

$$\nabla_{\mathbf{x}}\mathbf{p}_c = \mathbf{B}\mathbf{W}\hat{\mathbf{P}}_{\cdot c}, \tag{16}$$

Further, we define the Hessian as $\mathbf{H}$, where $\mathbf{H}_{ij}$ can be written as follows:

$$\begin{aligned}
\mathbf{H}_{ij} = \nabla_{\mathbf{x}_i}(\nabla_{\mathbf{x}_j}\mathbf{p}_c) &= \frac{\partial(\mathbf{B}_{j\cdot}\mathbf{W}\hat{\mathbf{P}}_{\cdot c})}{\partial \mathbf{x}_i} \\
&= \sum_{k=1}^{d} \frac{\partial(\mathbf{B}_{jk}\mathbf{W}_{k\cdot}\hat{\mathbf{P}}_{\cdot c})}{\partial \mathbf{x}_i} \\
&= \sum_{k \in \mathcal{K}} \frac{\partial(\mathbf{B}_{j,j+k}\mathbf{W}_{j+k,\cdot}\hat{\mathbf{P}}_{\cdot c})}{\partial \mathbf{x}_i} \\
&= \left(\sum_{k \in \mathcal{K}} \mathbf{B}_{j,j+k}\mathbf{W}_{j+k,\cdot}\right) \frac{\partial\hat{\mathbf{P}}_{\cdot c}}{\partial \mathbf{x}_i} \\
&= \left(\sum_{k \in \mathcal{K}} a_{j+k}\mathbf{g}_k\mathbf{W}_{j+k,\cdot}\right) \frac{\partial\hat{\mathbf{P}}_{\cdot c}}{\partial \mathbf{x}_i}
\end{aligned} \tag{17}$$

and

$$\frac{\partial \hat{\mathbf{P}}_{c'c}}{\partial \mathbf{x}_i} = \begin{cases} (1 - 2\mathbf{p}_c)\nabla_{\mathbf{x}_i}\mathbf{p}_c & \text{if } c' = c, \\ \mathbf{p}_c\nabla_{\mathbf{x}_i}\mathbf{p}_{c'} + \mathbf{p}_{c'}\nabla_{\mathbf{x}_i}\mathbf{p}_c & \text{otherwise}. \end{cases} \tag{18}$$

After computing the gradient and the Hessian, we now derive the decoy-enhanced saliency score $Z_j$ for $\mathbf{x}_j$, given a population of saliency scores $\tilde{E}_j = \{E(\tilde{\mathbf{x}}^1; F)_j, E(\tilde{\mathbf{x}}^2; F)_j, \cdots, E(\tilde{\mathbf{x}}^n; F)_j)\}$. According to Equation 3, $Z_j$ can always be decomposed into $Z_j = Z_j^+ + Z_j^-$ where $Z_j^+ = \max(\tilde{E}_j) - E(\mathbf{x}; F)_j$ and $Z_j^- = E(\mathbf{x}; F)_j - \min(\tilde{E}_j)$, respectively. Without loss of generality, we only focus on $Z_j^+$, noting that a smilar analysis is applicable to $Z_j^-$ as well.

Let $\tilde{\mathbf{x}} \in \{\tilde{\mathbf{x}}^1, \tilde{\mathbf{x}}^2, \cdots, \tilde{\mathbf{x}}^n\}$ denote the decoy which maximizes $E(\tilde{\mathbf{x}}; F)_j - E(\mathbf{x}; F)_j$.

The second-order Taylor expansion of the predicted $F^c(\mathbf{x})$ for target class $c$ around $\mathbf{x}$ is as follows:

$$F^c(\mathbf{x}) \approx F^c(\tilde{\mathbf{x}}) + \nabla_{\tilde{\mathbf{x}}}F^c(\tilde{\mathbf{x}})^T\Delta + \frac{1}{2}\Delta^T H_{\tilde{\mathbf{x}}}\Delta \tag{19}$$

where $\Delta = \mathbf{x} - \tilde{\mathbf{x}}$ and $H_{\tilde{\mathbf{x}}}$ is the Hessian of the neural network model on the decoy $\tilde{\mathbf{x}}$ whose entries are $(H_{\tilde{\mathbf{x}}})_{i,j} = \frac{\partial^2 F^c}{\partial \tilde{\mathbf{x}}_i \partial \tilde{\mathbf{x}}_j}$. By definition of the swappable condition in Equation 1, $F^c(\mathbf{x}) = F^c(\tilde{\mathbf{x}})$. Therefore, the following equation holds:

$$\nabla_{\tilde{\mathbf{x}}}F^c(\tilde{\mathbf{x}})^T\Delta \approx -\frac{1}{2}\Delta^T H_{\tilde{\mathbf{x}}}\Delta \tag{20}$$

Consider the case that the swappable patches are of size $1 \times 1$ at position $i$; then Equation 20 can be decomposed into

$$E(\tilde{\mathbf{x}}; F)_i = (\nabla_{\tilde{\mathbf{x}}}F^c(\tilde{\mathbf{x}}))_i \approx \frac{1}{2}(\tilde{\mathbf{x}} - \mathbf{x})_i(H_{\tilde{\mathbf{x}}})_{i,i}, \tag{21}$$

Then, according to Equation 21, we have

$$\begin{aligned} Z_j &= \frac{1}{2}(\tilde{\mathbf{x}} - \mathbf{x})_j(H_{\tilde{\mathbf{x}}})_{jj} - \nabla_{\mathbf{x}_j}\mathbf{p}_c \\ &\leq \left| \frac{1}{2}(\tilde{\mathbf{x}} - \mathbf{x})_j \sum_{k \in \mathcal{K}} \tilde{a}_{j+k}\mathbf{g}_k\mathbf{W}_{(j+k)\cdot}\tilde{\mathbf{P}}_{\cdot c} \right| + C_1 \\ &\leq C_2 \left|(\tilde{\mathbf{x}} - \mathbf{x})_j\right| \sum_{k \in \mathcal{K}} \tilde{a}_{j+k} + C_1 \end{aligned} \tag{22}$$

where $C_1 = \left|\nabla_{\mathbf{x}_j}\mathbf{p}_c\right|$ and $C_2 = \max_{k \in \mathcal{K}}\left|\mathbf{g}_k\mathbf{W}_{(j+k)\cdot}\tilde{\mathbf{P}}_{\cdot c}\right|$, respectively. Note that both $C_1$ and $C_2$ are linear combinations of the gradient, which is bounded by some Lipschitz constant (Szegedy et al., 2013).

## A8 SANITY CHECK FOR DECOY-ENHANCED SALIENCY MAPS

As suggested by Adebayo et al. (2018), any valid saliency methods should pass the sanity check in the sense that the saliency method should be dependent on the learned parameters of the predictive model, instead of edge or other generic feature detectors. We performed the model parameter randomization test (Adebayo et al., 2018) by comparing the output of the proposed saliency method on a pretrained VGG16 network with the output of the proposed saliency method on a weight-randomized VGG16 network. If the proposed saliency method indeed depends on the learned parameters of the model, it is expected that the outputs between the two cases differ substantially.

Following the cascading randomization strategy (Adebayo et al., 2018), the weights of pretrained VGG16 network are randomized from the top to bottom layers in a cascading fashion. This cascading randomization procedure is designed to destroy the learned weights successively. As illustrated in Figure A8, the cascading randomization destroys the decoy-enhanced saliency maps combined with three existing saliency methods, qualitatively. The conclusion is also supported by quantitative comparison measured by the structural similarity index (SSIM), shown in Figure A9.

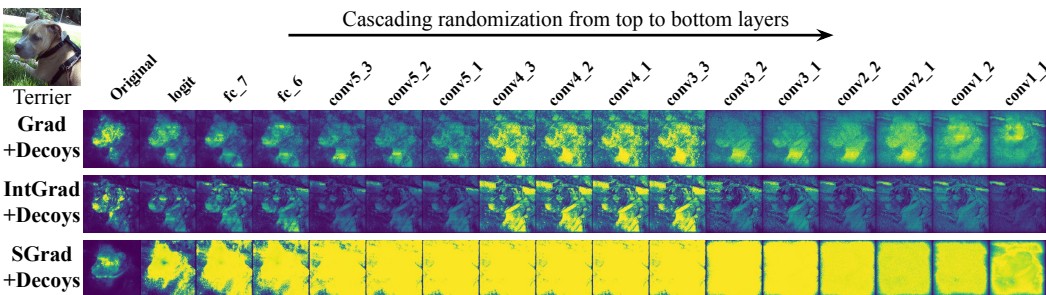

Figure A8: Cascading randomization on VGG16 network. The figure shows the original saliency map (first column) for the terrier. Progression from left to right corresponds to complete randomization of the pretrained VGG16 network weights from the top layer to the bottom layer.

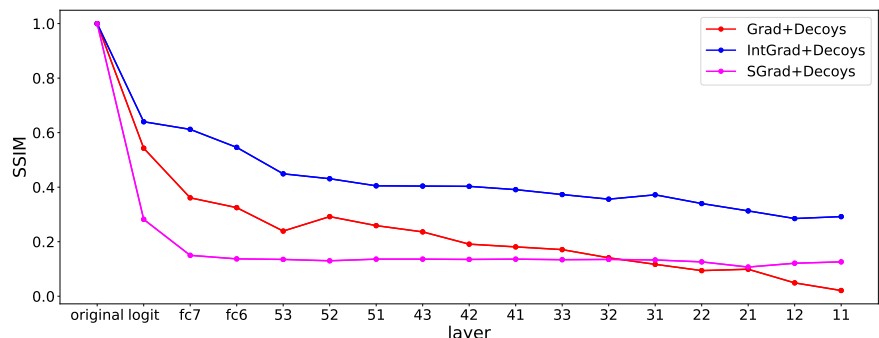

Figure A9: Structural similarity index (SSIM) for Cascading Randomization on VGG16 network.

## A9 SENSITIVITY OF HYPERPARAMETER SELECTION

The optimization formulated in Equation 6 involves one hyperparamer, $c$, the initial coefficient to weigh the intermediate representation difference between the decoy and the original image. Note that we use the same hyperparameter setting across all the images, rather than for each image separately. To evaluate how the selection of $c$ affects the performance, we carried out optimizations with respect to a wide range of $c \in \left\{ 10^1, 10^2, 10^3, 10^4, 10^5 \right\}$. As illustrated in Figure A10, the choice of initial coefficient c makes a negligible difference of three state-of-the-art saliency methods, both qualitatively and quantitatively.

## A10 RUN TIME OF DECOY OPTIMIZATION

To evaluate the computational cost of optimizing Equation 6, we carried out the run time comparison between optimizing one decoy and calculating three types of state-of-the-art saliency methods. The comparison is repeated 500 times with respect to different patch masks. As illustrated in Figure A11, on average optimizing one decoy takes 37.7% of the run time of the fastest, vanilla gradient-based saliency method. For other methods, the optimization is even less expensive, in a relative sense.

## A11 VALIDITY OF GENERATING DECOY BY BLURRING

As shown in Figure 3, combining blurry images with our decoy-enhanced saliency scores achieves empirically good performance even though the blurry images violate the design of decoys which resemble the intermediate representation of the original image. To understand why blurry images are empirically good, we compared the relative difference between the intermediate representation of the original images and the decoy/blurry images. Here, the relative difference is defined as the

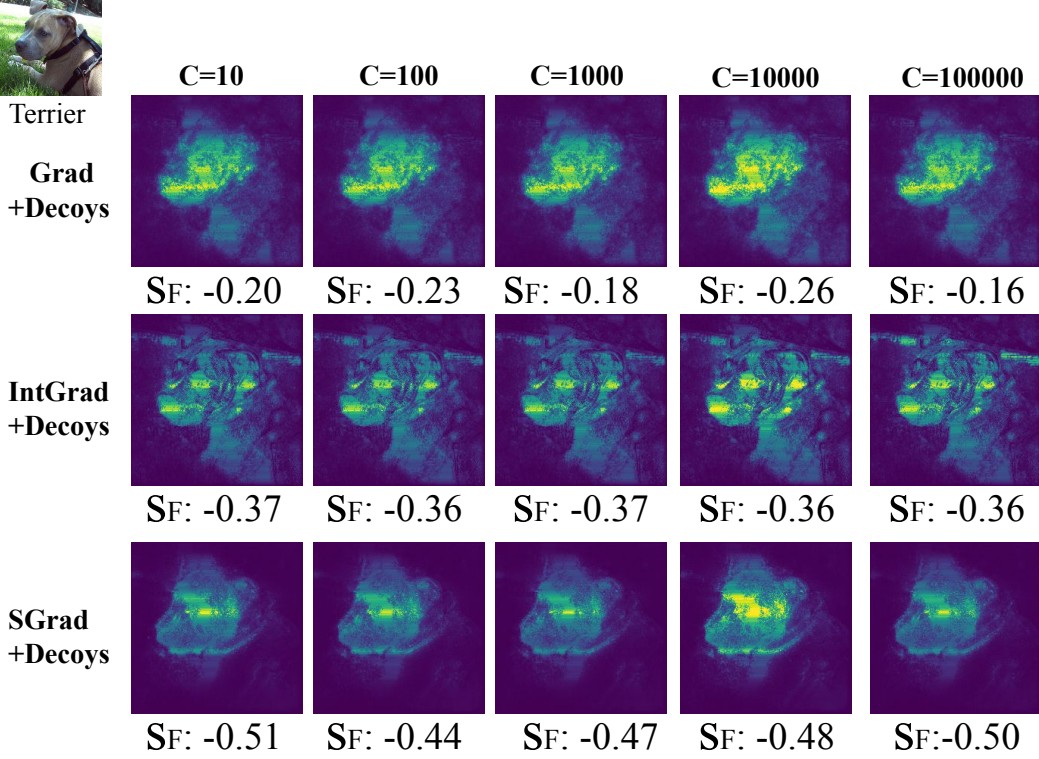

Figure A10: Visualization of saliency maps optimized using different hyperparamer $c$.

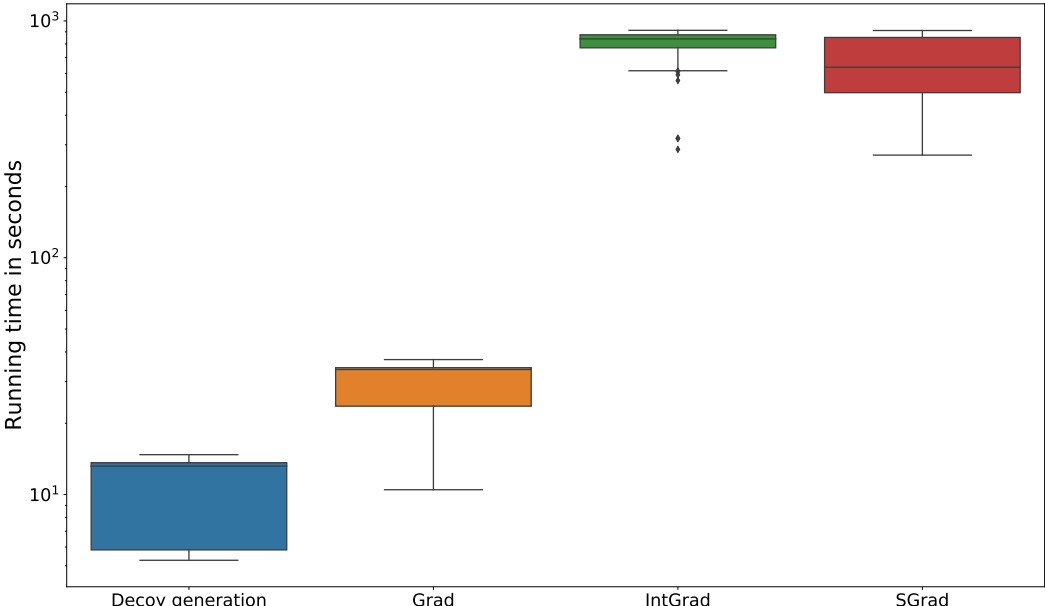

Figure A11: Run time to optimize one decoy and calculate three state-of-the-art saliency maps of the decoy. The comparison is conducted in the same CPU to guarantee fairness.

L-infinity norm between two intermediate representations divided by the maximum absolute value of any intermediate representation.

Because the decoy images are designed to preserve the intermediate representation of the original images, the relative difference is expected to be small. On the other hand, for the blurry images, the relative difference should be arbitrary since there is no such constraint. As illustrated in Figure A12, in the first maxpool layer (*i.e.,* a layer near the bottom of the network), the relative difference is large for blurry images (0.307 on average) and very small for decoy images (0.006 on average) as expected. However, in the last fully-connected layer (*i.e.,* a layer near the top of the network), the relative difference is much smaller for blurry images (0.034 on average) and remains small for decoy images (0.002 on average). In conclusion, even though the blurry images violate the constraint of decoys, this violation would be mitigated in deeper layers of the network.

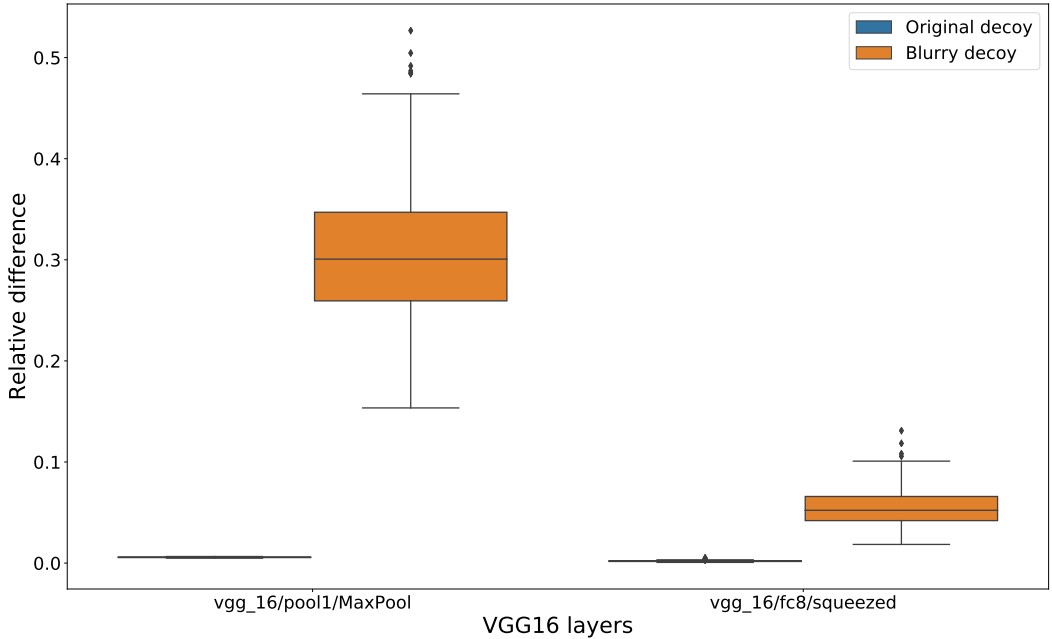

Figure A12: Relative difference between the intermediate representation of the original images and decoy/blurry images.

## A12 OBJECT LOCALIZATION

A standard method to evaluate saliency maps is by object localization (Dabkowski & Gal, 2017; Fong & Vedaldi, 2017), where the model was trained with the class label only without access to any localization data. We carried out Imagenet ILSVRC'14 localization task (Russakovsky et al., 2015) which contains 50K ImageNet validation images with annotated bounding boxes as ground truth. For each of the image in the dataset, we first calculated the gradient-based saliency maps with and without using blurry decoys, based on a pretrained VGG16 network. Following the preprocessing steps suggested by Dabkowski & Gal (2017); Fong & Vedaldi (2017), for each calculated saliency maps, a bounding box for the most dominant object is predicted by thresholding the corresponding saliency map and extracting the tightest bounding box that contains the whole remaining saliency map.

We investigated three different thresholding strategies suggested by Fong & Vedaldi (2017), including value thresholding, energy thresholding, and mean thresholding. Following the evaluation protocol suggested by Dabkowski & Gal (2017); Fong & Vedaldi (2017), the extracted localization box has to have Intersect over Union (IoU) greater than 0.5 with any of the ground truth bounding boxes in order to consider the localization successful, otherwise it is regarded as failure. As shown in Table A1, in terms of accuracy, decoy-enhanced saliency maps perform better than vanilla saliency maps without decoys.

| Accuracy | Value thresholding (0.25) | Energy thresholding (0.25) | Mean thresholding (0.25) |
|---|---|---|---|
| Gradient | 0.662 | 0.715 | 0.662 |
| Gradient+Decoys | **0.722** | **0.723** | **0.665** |

Table A1: ImageNet localization accuracy on VGG16 network using different thresholding strategies.

## A13 EXPERIMENTAL SETTINGS

| | $\ell$ | $c$ | $\epsilon$ | patch_size | stride | $\tau$ |
|---|---|---|---|---|---|---|
| AlexNet | 6 | 10000 | 200 | 3 | 1 | 1 |
| VGG16 | 3 | 10000 | 200 | 3 | 1 | 1 |
| ResNet | 2 | 10000 | 200 | 3 | 1 | 1 |

Table A2: The experimental settings of proposed methods for all the target models.

The experimental settings of the proposed approach on three convolutional neural networks (*i.e.,* AlexNet, VGG16, ResNet) are shown in Table A2. In the table, $\ell$ is the index of the layer within the target model that is selected to generate the decoy images. The hyperparameter $c$ controls the weight of $\|F_\ell(\tilde{\mathbf{x}}) - F_\ell(\mathbf{x})\|_\infty$, and $\epsilon$ refers to the tolarence of $\|F_\ell(\tilde{\mathbf{x}}) - F_\ell(\mathbf{x})\|_\infty$ in Equation (6). The patch_size and stride control the size and the stride step of each decoy patch.

## A14 ADDITIONAL EXPERIMENTAL RESULTS

Figure A13–A16 provide more examples of the fidelity, stability and generalizability of the decoy-enhanced saliency methods. These results are consistent with those shown in the Figures 3–6.

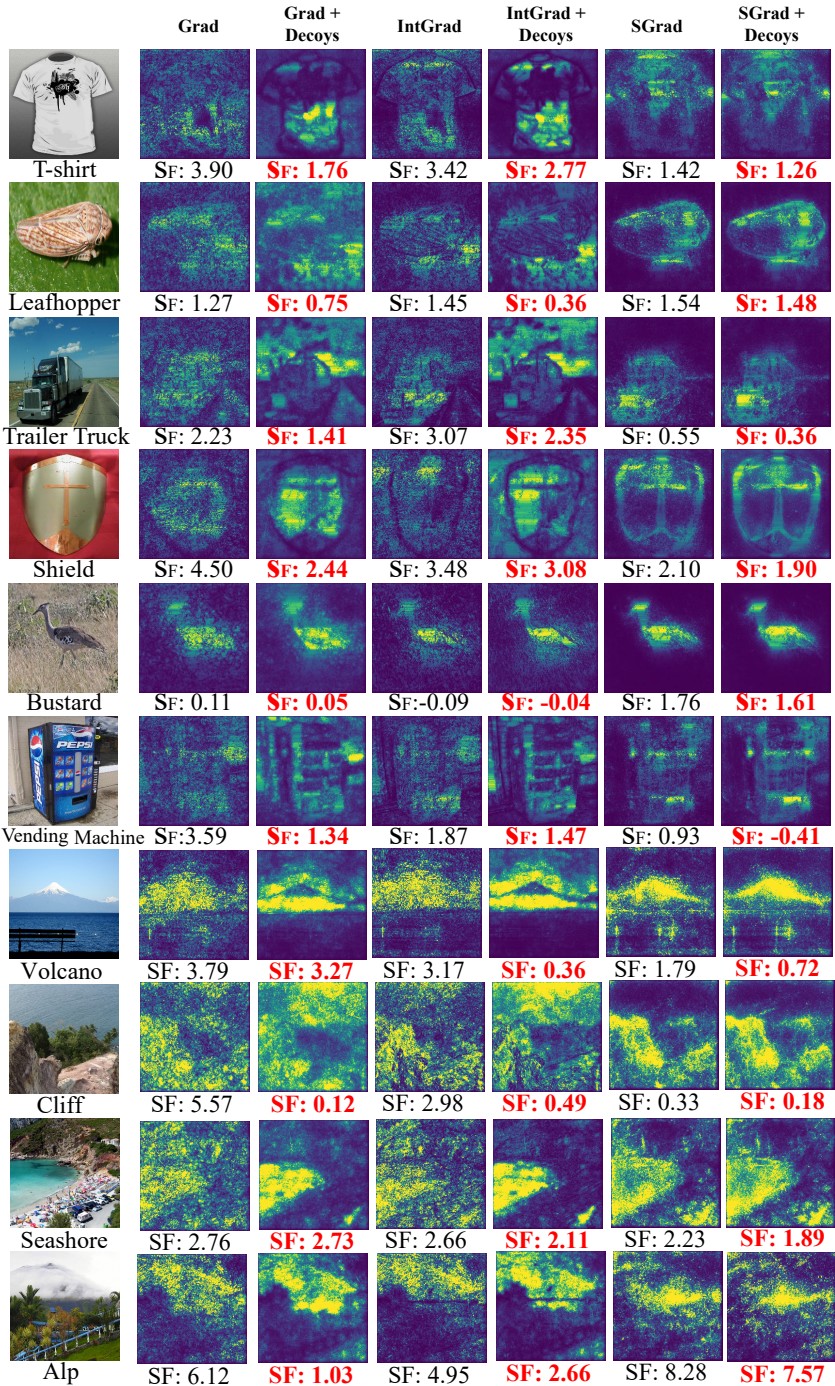

Figure A13: Fidelity evaluation results of four more images on VGG16.

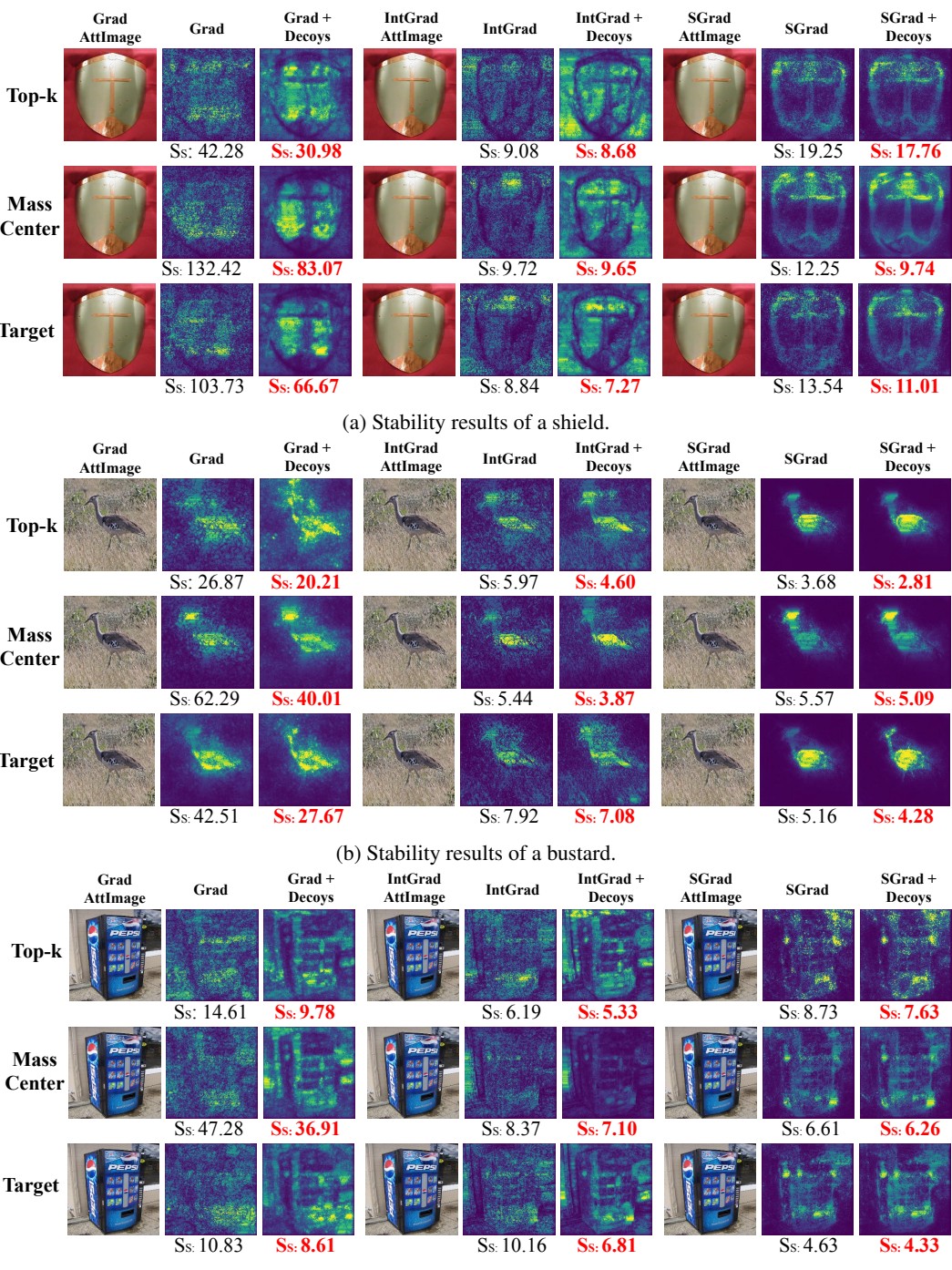

(a) Stability results of a shield.

(b) Stability results of a bustard.

(c) Stability results of a vending machine.

Figure A14: Stability evaluation results of three more images on VGG16.

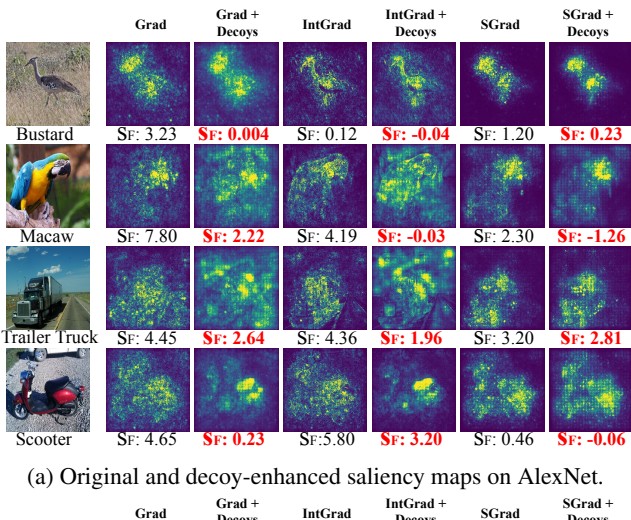

(a) Original and decoy-enhanced saliency maps on AlexNet.

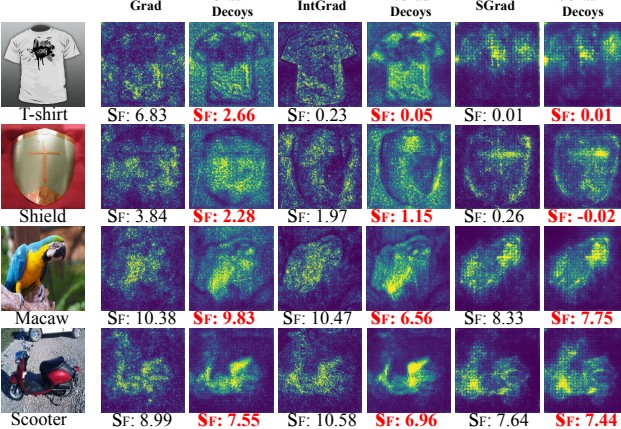

(b) Original and decoy-enhanced saliency maps on ResNet.

Figure A15: The illustration of the performances of the proposed approach of more images on AlexNet and ResNet.

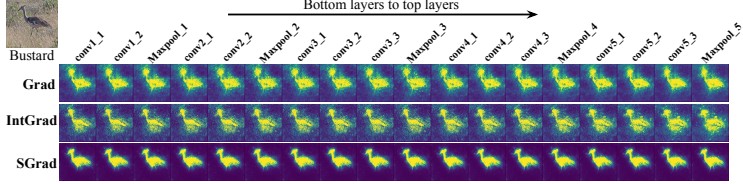

(a) The mean and standard derivation of $S_F$ score for gradient, integrated gradient and SmoothGrad are: (-0.01, 0.01), (-0.03, 0.01), (0.09, 0.01).

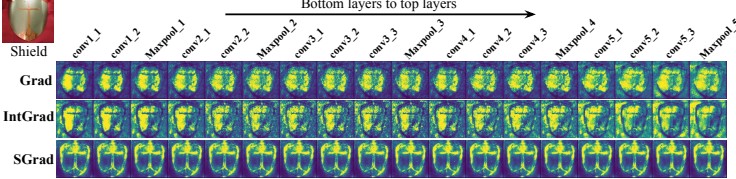

(b) The mean and standard derivation of $S_F$ score for gradient, integrated gradient and SmoothGrad are: (3.34, 0.73), (2.21, 1.33), (1.49, 0.18).

Figure A16: Demonstrations of decoy-enhanced saliency maps generated from each convolutional and pooling layer in VGG16.

