# OpenReview forum: "Robust saliency maps with distribution-preserving decoys"
_ICLR.cc/2020/Conference — Reject_

### Official Review · AnonReviewer1 · 2019-10-18
**Official Blind Review #1**

**Rating:** 3

**Review:**

The authors tackle the important problem of generating saliency maps. First, a decoy image is defined (in short, it is a perturbed version of the image that leads to very similar activations) and then, the method that leverages decoy images is proposed. The method can be understood as a improvement that can be applied to enhance an existing saliency extractor. For a given image, the method generates a bunch of decoys images, and then a given saliency extractor is applied to not only the original image but all generated decoy images. The resulting saliency maps are aggregated to output the final saliency map. I found this idea technically sound.

However, there are two keys reasons why this paper should be rejected.
(1) The idea of using decoy images is interesting but computationally expensive. In practice (as showed in the paper) using blurred images leads to comparable results.
(2) The paper misses one very important experiment which is a quantitative comparison with the existing works on Imagenet ILSVRC’14 localization task. This is the standard experiment and is used in a few works cited (Fong & Vedaldi, 2017; Dabkowski & Gal, 2017).

Even though the idea of using decoy images is technically sound, I find the algorithm for generating these decoy images very complex and computationally expensive. First, under Eq.6 one can find "Our strategy is to set c to a small value initially and run the optimization. If it fails, then we double c and repeat until success." And then below Eq.8 there is "After each iteration, if the second term in Equation 8 is zero, indicating that τ is too large, then we reduce τ by a factor of 0.95 and repeat". Hence, the algorithm is nested and the paper does not provide any details how long the procedure is in practice. On top of that, since a population of decoy images is needed (12 in the experiments), this expensive procedure has to be run a few times. When decoy images are replaced with blurry images the results are almost the same and hence, unfortunately, using decoy images is not justified.

In general, the experimental results are decent and prove the possible benefits of using the population of images (decoys or blurred ones). However, I think that Imagenet ILSVRC’14 localization task is the standard experiment that provides the quantitative view and should be performed. The paper evaluates on Imagenet data set already, and hence adding this experiment should be straightforward.

The paper does a good job motivating the problem and covering related work. The paper states that the key limitation of existing saliency maps is that they "evaluate pixel-wise importance in an isolated fashion by design, implicitly assuming that other pixels are fixed" and "the presence of gradient saturation". While it is valid for gradient-based methods, it is not for perturbation-based methods. The latter are just briefly mentioned and then it is stated that "for any perturbation-based method, a key challenge is ensuring that the perturbations are effective yet preserving the training distribution". However, I do not find this reason to be strong enough to exclude these work from consideration, because there are works that train saliency extractor and the classifier simultaneously and then the argument mentioned does not hold (Fan et al., 2017; Zolna et al., 2018).

There is one more thing that I would like to ask about. In Eq.3, Z_j is defined as max(E˜_j ) − min(E˜_j ). Hence, if a given pixel is very important for all decoy images, all elements of E˜_j will be high and then Z_j will be very small. As a result, a saliency value assigned to this pixel will be low which seems to be counter-intuitive. Can you please elaborate on that?


(Fong & Vedaldi, 2017): Ruth C Fong, and Andrea Vedaldi. Interpretable explanations of black boxes by meaningful perturbation
(Dabkowski & Gal, 2017): Piotr Dabkowski, and Yarin Gal. Real time image saliency for black box classifiers
(Fan et al 2017): Lijie Fan, Shengjia Zhao, and Stefano Ermon. Adversarial localization network
(Zolna et al 2018): Konrad Zolna, Krzysztof J. Geras, and Kyunghyun Cho. Classifier-agnostic saliency map extraction

**Experience Assessment:**

I have published one or two papers in this area.

**Review Assessment: Checking Correctness Of Derivations And Theory:**

N/A

**Review Assessment: Checking Correctness Of Experiments:**

I carefully checked the experiments.

**Review Assessment: Thoroughness In Paper Reading:**

I read the paper thoroughly.

---

> ### Author Response · Authors · 2019-11-13
> **All comments by the reviewer have been addressed**
>
> We thank the reviewer for the feedback. Please see below for our clarification.
>
> The reviewer questioned about the computational efficiency in calculating decoy images. We carried out a run time comparison between optimizing one decoy and calculating three types of saliency maps. We repeated this comparison 500 times w.r.t different patch masks.  The results (Figure A11) show that on average optimizing one decoy takes  37.7% of the run time of the fastest, gradient-based saliency method. For other methods, the optimization is even less expensive, in a relative sense.
>
> The reviewer questioned the necessity of using decoy images because blurred images lead to comparable results. To understand why blurry images are empirically good, we compared the relative difference between the intermediate representation of the original images and the decoy/blurry images (Appendix section A11). The relative difference of decoy images is expected to be small by design whereas the relative difference of blurry images should be arbitrary since there is no such constraint. The results (Figure A12) show that, in the first maxpool layer, the relative difference is large for blurry images (0.307 on average) and very small for decoy images (0.006 on average) as expected. However, in the last fully-connected layer, the relative difference is much smaller for blurry images (0.034 on average) and remains small for decoy images (0.002 on average). In conclusion, even though the blurry images violate the constraint of decoys, this violation is mitigated in deeper layers of the network. From the practitioner’s perspective, blurry images can be used as more efficient alternatives. However, decoy images are still necessary to justify the theoretical soundness of the decoy-enhanced saliency score.
>
> The reviewer suggested a quantitative comparison on Imagenet ILSVRC’14 localization task. We carried out the localization task which contains 50K ImageNet validation images with annotated bounding boxes as ground truth. For each of the images in the dataset, we first calculated the gradient-based saliency maps with and without using blurry decoys, based on a pretrained VGG16 network. The tightest bounding box bounding box for each saliency map is extracted after thresholding [1,2]. The extracted localization box has to have IoU>0.5 in order to consider the localization successful, failure otherwise. The results (see Table A1) show that, in terms of accuracy, decoy-enhanced saliency maps perform better than vanilla saliency maps without decoys. We added text (see Appendix section A12) highlighting the quantitative comparison with the existing works on the Imagenet ILSVRC’14 localization task.
>
> The reviewer pointed out that some perturbation-based methods have made efforts to preserve the training distribution. We now cite the recommended papers in the revision. In addition, in response to a point raised by Reviewer #3, we added text (see Section 1, paragraph 4) highlighting the relationship between the proposed saliency method and counterfactual-based methods, which involve comparing against the closest image on the other side of the decision boundary
>
> The reviewer asked about the case where, “if a given pixel is very important for all decoy images, all elements of \tilde{E}_j will be high and then Z_j will be very small. As a result, a saliency value assigned to this pixel will be low which seems to be counter-intuitive. ” First, we would like to point out  that in a normal situation (i.e., when the image doesn’t suffer from an adversarial attack), an important pixel is not important in an isolated fashion. Instead, the important pixel tends to contribute a strong joint effect in conjunction with neighboring important pixels, to potentially capture meaningful patterns such as edges, texture, etc. In light of this observation, this particular important pixel will have more room to fluctuate without influencing the joint effect on the prediction. In such a case, some elements of \tilde{E}_j will be high and others will be low, contributing a large Z_j. On the other hand, in the unusual situation when an isolated important pixel is indeed observed, we tend to believe that the pixel has been adversarially attacked. As pointed out by the reviewer, the proposed decoy-enhanced saliency score Z_j will be low, which is what we want. The special case raised by the reviewer helps to explain the reason why the proposed method is robust to adversarial attack. We added text (see Section 4.3, paragraph 4) highlighting this special case as an explanation of why our method is robust to adversarial attacks.
>
> References:
> [1] Fong, Ruth C., and Andrea Vedaldi. "Interpretable explanations of black boxes by meaningful perturbation." Proceedings of the IEEE International Conference on Computer Vision. 2017.
> [2] Dabkowski, Piotr, and Yarin Gal. "Real time image saliency for black box classifiers." Advances in Neural Information Processing Systems. 2017.

---

> > ### Comment · AnonReviewer1 · 2019-11-15
> > **Reply**
> >
> > I have read authors' response and relevant updates in the paper.
> >
> > I admit that authors have done good job and tried to address all my comments. Two the most important experiments conducted are a) ablation study on blurry images and b) quantitative comparison on the Imagenet ILSVRC'14 localization task.
> >
> > Unfortunately, the results obtained with extra experiments are not convincing. Actually, authors agreed that using blurred images leads to comparable results and only show why it is the case. Hence, I still do not understand the motivation behind using decoy images in practice.
> >
> > The improvement on the localization task is marginal, especially for the best performing thresholding methods.
> >
> > Overall, I find the paper interesting but I think it does not meet the bar for ICLR conference.

---

> > > ### Author Response · Authors · 2019-11-15
> > > **Response to reviewer #1**
> > >
> > > We would like to thank the reviewer for responding our response. Please allow us to make some clarifications below.
> > >
> > > 1. We would like to highlight the most important contribution of the paper, the decoy-enhanced saliency score. Essentially, we derived a robust saliency measure to provably address the two key limitations of existing gradient-based saliency maps, "evaluate pixel-wise importance in an isolated fashion” and "the presence of gradient saturation", with theoretical guarantees, which matches well with the interests of ICLR.
> > >
> > > 2. The reviewer said they “still do not understand the motivation behind using decoy images in practice.” As we pointed out in the previous response, blurry images can indeed be used to replace decoys in practice. In particular, we show that the decoy-enhanced saliency score works well with blurry images empirically, despite the lack of theoretical guarantees. However, from a theoretical perspective, the definition of decoys is necessary to justify the theoretical soundness of the decoy-enhanced saliency score.
> > >
> > > 3. The reviewer pointed out that “the improvement on the localization task is marginal. ” While the reviewer may find the 9.1% increase in accuracy between vanilla saliency maps with and without decoys to be small, we would like to draw their attention to the consistency of the improvement across various tasks. Since there is no existing method that provably compensates for gradient saturation and takes into account joint activation patterns, any effective method that achieves these goals is significant. We believe that the insights offered by our decoy-enhanced saliency score will inspire further efforts to develop theoretically sound methods to interpret DNNs.

---

### Official Review · AnonReviewer3 · 2019-10-24
**Official Blind Review #3**

**Rating:** 3

**Review:**

Given a model and an image, the proposed method generates perturbations of the image, such that the model output in intermediate layers of the network does not change. Their method first generates such perturbed images, and then uses those to generate a saliency map in order to interpret the classification.  The idea is overall interesting, but its relationship with many other methods is not adequately examined. Significant portion of recent literature is ignored, especially studies on counterfactuals and decision boundaries in the context of image classification.  For numerical results, they compare their results with raw scores generated by three other methods, but the results are quite thin and disorganized with respect to adversarial robustness and interpretability.

Your proposed computational method possibly needs to be changed, too.   Details are explained below:

Authors have cited the paper on "Sanity Checks for Saliency Maps", but have not used it to verify their own results, not with respect to saliency map, and not with respect to computational cost.  Several recent methods on saliency maps have not been considered, for example:
1. Certifiably robust interpretation in deep learning
2. Interpreting Neural Networks Using Flip Points
3. Explaining Image Classifiers by Counterfactual Generation
 4. Counterfactual Visual Explanations

It's not clear how your proposed saliency scores relate to methods that consider counterfactuals and the decision boundaries. For any image, there is a counterfactual image closest to it. One can also compute the closest image on the decision boundary of the model. Such image can reveal which pixels should be changed in order to change or to keep the classification. The relationship between the current method and those methods needs to be established.
 Assume that for an image, you compute the closest image to it on the decision boundary and let’s call the distance between them r (distance measured in l1 or l2 norm). Then, consider a ball centered at that image with radius r. All images inside that ball would have the same label as the original image and you can obtain all of them only by solving one optimization problem for finding closest point on the decision boundary. Would this be a less expensive computation compared to solving your proposed decoy optimization many times?

There are many hyper parameters in your optimization problem, and it seems that for each image, hyper-parameters should be tuned separately. However, if you seek the closest image on the decision boundary, your results would be independent of hyper-parameters.

As you have mentioned, your optimization problem seems quite hard to solve, but no details are given on how long it takes to solve it for a single image. You have mentioned sometimes you cannot solve it if c is not chosen well. This information needs to be reported. Is it solvable at all with a large mask? How many trials did it take to obtain your saliency maps?

Your optimization is nonlinear, non-convex, non-differentiable, …. . How did you deal with non-convexity? Do you arrive at the same solution each time you solve it? These are vital information to about an optimization method.

Proposition 1 is quite obvious. Its discussion seems obvious, too, given how you have defined the Z. In your proof, you have considered a single layer network, i.e. a simple linear transformation. It might be better to explain your proof for Proposition 1 as an observation and use it as the motivation on how you have defined Z.
 Overall, you are taking a data-driven approach to interpret the classification of an image. The additional computational cost of your proposed method needs to be compared to other methods that are not data-driven. For example, how does your computational cost compare with the references above?

On page 12, one equation is referenced with ??.

At the beginning of section 3.4, you are using a standard optimization technique to augment a constraint as a penalty in the objective function. You can explain it as such, instead of saying we solve an alternative formulation. See Numerical Optimization by Nocedal and Wright.

The necessity to transform the variable via tanh, instead of performing projection is not explained. Does it make the process faster?

Strategy for choosing a good initial value for c (in the penalty term) and how to increase it is not explained well. How many iterations does it take to find the solution?

Any classification model is defined by its decision boundaries, so interpreting the classification without considering the decision boundaries seems inconclusive.

In summary, it would be best to compare your results with other methods, some of which are mentioned above. It would also be necessary to justify the additional data-driven cost that you are prescribing and to explain why your goals cannot be achieved otherwise with less expensive computation.




**Experience Assessment:**

I have published one or two papers in this area.

**Review Assessment: Checking Correctness Of Derivations And Theory:**

I carefully checked the derivations and theory.

**Review Assessment: Checking Correctness Of Experiments:**

I assessed the sensibility of the experiments.

**Review Assessment: Thoroughness In Paper Reading:**

I read the paper thoroughly.

---

> ### Author Response · Authors · 2019-11-13
> **All comments by the reviewer have been addressed**
>
> We thank the reviewer for the feedback. Please see below for our clarification.
>
> The reviewer asked about the sanity check on our method. Please refer to the response to review #2 for the detailed explanation.
>
> The reviewer pointed out four missing citations. We now add them in the revision.
>
> The reviewer asked about the relationship between the proposed method and counterfactual-based methods. The major differences are briefly threefold: (1) counterfactual images seek a minimum set of features to exclude or include w.r.t the prediction score [1] whereas saliency maps characterize the influence of each feature; (2) counterfactual images are optimized toward the aforementioned objective w.r.t the decision boundary whereas decoys are independent of labels, without changing the decision boundary. (3) counterfactual images potentially could be out-of-distribution whereas decoys don’t by design, We added more details (Section 1, paragraph 4) highlighting the relationships described above.
>
> The reviewer asked about the hyperparameter tuning strategy. Our method only introduce one hyperparameter: c. We use the same hyperparameter setting across all the images, rather than for each image separately. We carried out the optimization w.r.t a wide range of c, ranging from 10 to 100000. The results (Figure A10) show that the choice of c makes a negligible difference, both qualitatively and quantitatively.
>
> The reviewer questioned the effect of choosing improper c. First, we are following a standard strategy for optimizing adversarial examples [2]. Second, as described in the previous paragraph, the choice of c makes a negligible difference. Finally, though the optimization may fail if c hits a large upper bound (10^10 in our case), we never encountered any failures throughout our experiments. Therefore, we empirically start with a large initial coefficient c (i.e. c=10000), and double it each time. This procedure takes at most dozens of iterations to find the solution.
>
> The reviewer asked about the runtime to solve the optimization problem. We carried out a run time comparison between optimizing one decoy and calculating three types of saliency maps. We repeated this comparison 500 times w.r.t different patch masks.  The results (Figure A11) show that on average optimizing one decoy takes  37.7% of the run time of the fastest, gradient-based saliency method. For other methods, the optimization is even less expensive, in a relative sense.
>
> The reviewer is correct that the optimization is “nonlinear, non-convex and non-differentiable.” This is why we solved an alternative formulation (Equation 6) suggested by [2], which is nonlinear, non-convex, but differentiable. Similar to any deep learning optimization, the optimization converges to different local minima each time it is run; however, the results (Figure A10) show that empirically the optimization ends up with quite consistent solutions.
>
> The reviewer claims that “Proposition 1 is quite obvious.” While we appreciate that the reviewer may find this proposition obvious, we prefer to retain the proof in the supplement for the benefit of less expert readers, who may not find it obvious. The reviewer also states that we “have considered a single layer network, i.e. a simple linear transformation.” However, our single-layer network contains both a ReLU unit and a softmax, so is not equivalent to a linear transformation.  Finally, the reviewer suggests that we “explain your proof for Proposition 1 as an observation and use it as the motivation on how you have defined Z.” We have made this change into the discussion (Section 3.4, paragraph 2)
>
> The reviewer questioned the necessity of tanh transformation during the optimization. For this particular point, we followed the approach in [2]. We have amended the text (Section 3.4, paragraph 3) to point out other possibilities.
>
> The reviewer pointed out that “interpreting the classification without considering the decision boundaries seems inconclusive.” We agree that interpreting a classification requires considering the decision boundary, either implicitly or explicitly. Counterfactual-based methods involve comparing against the closest image on the other side of the decision boundary, thereby explicitly considering the decision boundary. In contrast, saliency methods consider the decision boundaries implicitly, by calculating the variants of the gradient w.r.t the label. Similarly, despite not explicitly considering the decision boundary during decoy construction, our proposed method implicitly considers the decision boundary when calling off-the-shelf saliency methods. We added detailed explanation (Section 1, paragraph 4) highlighting the relationship to the decision boundaries for both types of methods.
>
> References:
> [1] Dabkowski, Piotr, and Yarin Gal. "Real time image saliency for black box classifiers." NIPS. 2017.
> [2] Carlini, Nicholas, and David Wagner. "Towards evaluating the robustness of neural networks."  S&P, 2017.

---

### Official Review · AnonReviewer2 · 2019-10-29
**Official Blind Review #2**

**Rating:** 6

**Review:**

I Summary
This paper has two major contributions:
- A method to infer robust saliency maps using distribution preserving decoys (generated perturbated images that resemble the intermediate representation of the original image in the neural network)
- A decoy-enhanced saliency score which compensates for gradient saturation and takes into account joint activation patterns
The authors show the performance of their methods on three different saliency methods, quantitatively and qualitatively, but also when it is submitted to adversarial perturbations.

II Comments
1. Content
The paper is very clear and easy to read, I would like to point out how well structured it is.
The method yields very interesting results, the whole work is very complete experimentally (saliency methods used, adversarial perturbations, models etc) and the process coherent. I would love to see it applied to different datasets: Imagenet focuses more on the foreground, whereas "Places" focuses more on background objects and could be interesting to use.
The authors referred to Sanity Checks for Saliency Maps (Adebayo et al) without using it for their results, it would be nice to add it to the experiments.


2. Typos
- intro, paragraph 2, l2: use -> uses
- paragraph 5, l9: indendently ->  independently
- 4.2, paragraph 4, last line: target word twice
- 4.3, paragraph 2, l5: fisrt -> first
- l9: not able -> may not be able
- 4.5, l5: salinency -> saliency



**Experience Assessment:**

I have read many papers in this area.

**Review Assessment: Checking Correctness Of Derivations And Theory:**

I assessed the sensibility of the derivations and theory.

**Review Assessment: Checking Correctness Of Experiments:**

I assessed the sensibility of the experiments.

**Review Assessment: Thoroughness In Paper Reading:**

I read the paper at least twice and used my best judgement in assessing the paper.

---

> ### Author Response · Authors · 2019-11-13
> **All comments by the reviewer have been addressed**
>
> We would like to thank the reviewer for reviewing our paper. Please allow us to make some clarifications below.
>
> The reviewer pointed out that “I would love to see it applied to different datasets: Imagenet focuses more on the foreground, whereas ‘Places’ focuses more on background objects and could be interesting to use.” We are unaware of a dataset that focuses on the background objects only, so instead we investigated the images in ImageNet which contain background objects, such as volcano, cliff, and seashore. The results (see Figure A13) show that decoys consistently help to produce more visually coherent and quantitatively better saliency maps.
>
> The reviewer pointed out that “The authors referred to Sanity Checks for Saliency Maps (Adebayo et al) without using it for their results, it would be nice to add it to the experiments.” First, in our paper, we limit our focus to three saliency methods that pass the sanity check [1]. Second, we carried out the model parameter randomization test [1] by comparing the output of the proposed saliency method on a pretrained VGG16 network with the output of the proposed saliency method on a VGG16 network that has been randomized from the top to bottom layers in a cascading fashion. The results (see Figure A8 and A9) show that the outputs between the two cases differ substantially, both qualitatively by visualization and quantitatively by the structural similarity index (SSIM). This observation suggests that our proposed saliency method does not violate the sanity check [1].
>
> References:
> [1] Adebayo, Julius, et al. "Sanity checks for saliency maps." Advances in Neural Information Processing Systems. 2018.

---

### Decision · Program_Chairs · 2019-12-19

**Decision:**

Reject

**Comment:**

This submission proposes a method to explain deep vision models using saliency maps that are robust to certain input perturbations.

Strengths:
-The paper is clear and well-written.
-The approach is interesting.

Weaknesses:
-The motivation and formulation of the approach (e.g. coherence vs explanation and the use of decoys) was not convincing.
-The validation needs additional experiments and comparisons to recent works.

These weaknesses were not sufficiently addressed in the discussion phase. AC agrees with the majority recommendation to reject.